# Cost-effectiveness analysis of durvalumab as a maintenance treatment for patients with locally advanced, unresectable, stage III nsclc in china

Xiaotong Jiang[1], Jinyu Chen [2,3]*, Min Zheng[4], Hanxue Jia[4]

1 Institute of Medical Information, Chinese Academy of Medical Sciences, Beijing, China, 2 China Health Economics Association, Beijing, China, 3 National Center for Medicine and Health Technology Assessment, Beijing, China, 4 Shenyang Pharmaceutical University, Shenyang, China

* sycjy1015@163.com

## Abstract

### Objective

The aim of this study was to evaluate the cost-effectiveness of durvalumab compared with Best supportive care (BSC) after chemoradiotherapy in patients with stage III non-small cell lung cancer from healthcare system perspective in China.

### Methods

A dynamic state transition model was adopted to simulate life time, direct medical costs and QALYs. In the base case scenario, for patients with unresectable, stage III non-small cell lung cancer whose disease has not progressed after platinum-based chemoradiation therapy, the treatment group would use durvalumab whereas the control group would use BSC. Clinical data and health utility were derived from the patient-level data of Asian ethnicity in the PACIFIC trial. Cost of drug acquisition, follow-up, medical service, inspection, terminal care and adverse event treatment were considered in this model. The cost of durvalumab was calculated based on retail prices and Patient Assistance Program.

### Results

In the base case, the durvalumab group yielded an additional 2.60 LYs and 2.37QALYs (discounted), causing an additional cost of 0.459 million RMB and 0.109 million RMB without and with PAP, so the ICER was 193,898 RMB/QALY and 46,093.12 RMB/QALY respectively.

### Conclusions

This study demonstrated that durvalumab can improve the survival of patients with unresectable, stage III non-small cell lung cancer whose disease has not progressed after platinum-based chemoradiation therapy and would be a cost-effective option compared with

**Data Availability Statement:** All relevant data are within the manuscript and its Supporting Information files.

**Funding:** The author(s) received no specific funding for this work.

**Competing interests:** The authors have declared that no competing interests exist.

BSC at a willingness to pay (WTP) threshold of 212676 RMB (three times GDP per capita of China in 2019).

## Introduction

In 2015, there were about 787,000 new lung cancer cases in China, accounting for 20.03% of all new cancer cases; the number of lung cancer deaths reached up to 631,000 which represented 26.99% of the total number of cancer deaths [1]. Lung cancer has become a malignant tumor with the highest morbidity and mortality in China. Among which non-small cell lung cancer (NSCLC) represents the most common pathological type of lung cancer, approximately making up 85% of the total number of lung cancer [2, 3]. Stage III NSCLC, also called locally advanced NSCLC, accounts for 30% of NSCLC patients [4]. The treatment of stage III NSCLC has evolved from radiotherapy alone in the 1980s to concurrent chemoradiotherapy today, the 5-year survival rate, however, is still only 15% to 20% [5]. Emergence of immune checkpoint inhibitors in recent years has brought new hope to patients with stage III NSCLC. Durvalumab injection (hereinafter referred to as Durvalumab), the first approved PD-L1 inhibitor in China, is a fully humanized IgG1 monoclonal antibody against PD-L1 which has been recommended as a consolidation therapy following concurrent chemoradiotherapy for patients with unresectable stage III NSCLC by National Comprehensive Cancer Network (NCCN) guidelines, European Society for Medical Oncology (ESMO) guidelines and the Chinese Society of Clinical Oncology (CSCO) guidelines [6–8].

For the efficacy and safety of durvalumab, a phase III, randomized, double-blind, placebo-controlled international multicenter clinical study (NCT02125461, N = 713), also known as PACIFIC, was conducted to evaluate the treatment effect of durvalumab in patients with unresectable stage III NSCLC whose disease has not progressed after platinum-based, concurrent chemoradiation therapy. The results showed that durvalumab significantly prolonged the progression-free survival (17.2 months vs 5.6 months, HR = 0.51) and overall survival (not reached vs. 29.1 months, HR = 0.68); 3-year survival rate of patients in the durvalumab group was 57.0% which is 13.5% higher than that of patients in the placebo group [9, 10]. Safety data showed that the use of durvalumab in addition to definitive chemoradiation therapy did not significantly increase the incidence of adverse reaction events. To date, there are 6 studies on economic evaluation of durvalumab conducted out of China, which might be different in the study perspectives, models and resources of cost data, thus varied in conclusions [11–16]. No studies on cost-effectiveness of durvalumab has been found in China yet, therefore, the purpose of this study is to evaluate the cost-effectiveness of durvalumab versus best supportive care (BSC) for patients with unresectable, stage III NSCLC in China based on the cost data of in China and the effectiveness data from subpopulation analysis of Asian ethnicity in PACIFIC study (January 31, 2019, data cut-off), so as to provide reference for the dynamic adjustment of Chinese medical insurance catalogues and the negotiation of market exclusivity for durvalumab.

## Material and methods

### Target population

The target population in the model was unresectable stage III NSCLC patients who had not progressed after platinum-based, concurrent chemoradiation therapy. Patients in the treatment group were intravenously infused with durvalumab at a dose of 10 mg/kg once every 2 weeks, while those in the control group were treated with BSC at the same dose and frequency. The treatment duration of both groups were 12 months or until disease progression.

## Model structure

A dynamic state transition model was constructed in Microsoft ® Excel 2019 to analyse the clinical and economic outcomes of durvalumab and BSC from the perspective of Chinese healthcare system. The structure of Markov model consisted of three states: progression free (PF), progressed disease (PD) and death (DEATH) (Fig 1). The initial age of population in the simulated cohorts was set as 62.9 years old according to the average age of the Asia ethnicity in the PACIFIC study, and the initial state was PF. In addition, according to the dosing frequency and duration of PACIFIC study, the cycle period for the first 12 months in the model was set as 14 days, and then 28 days after 12 months. During each cycle, patients with a certain state would receive treatment with corresponding drugs. In order to fully demonstrate the benefits of these two treatment groups, the time horizon of this study was about 40 years which approximated a life time, i.e., the simulations of model will stop only when death state is presented for the modelled population of all cohorts. The therapeutic pathways of both groups were established in accordance with PACIFIC trial and recommendations of "2020 CSCO Guidelines for Diagnosis and Treatment of Non-small Cell Lung Cancer" as follows: following concurrent chemoradiation therapy with platinum-based regimen in the setting of PF state (at least 2 cycles), durvalumab or placebo will be administrated respectively until PD (up to 12 months) if PF remained; upon PD, subsequent treatment such as immunotherapy, targeted therapy and chemoradiotherapy will be given until death [8, 9].

## Clinical data

The clinical efficacy data of both treatment groups were obtained from the patient-level data of the Asian ethnicity sub-population in the PACIFIC study which enrolled a total of 192 Asian patients, with 120 assigned to the durvalumab group and 72 to the BSC group. In the model,

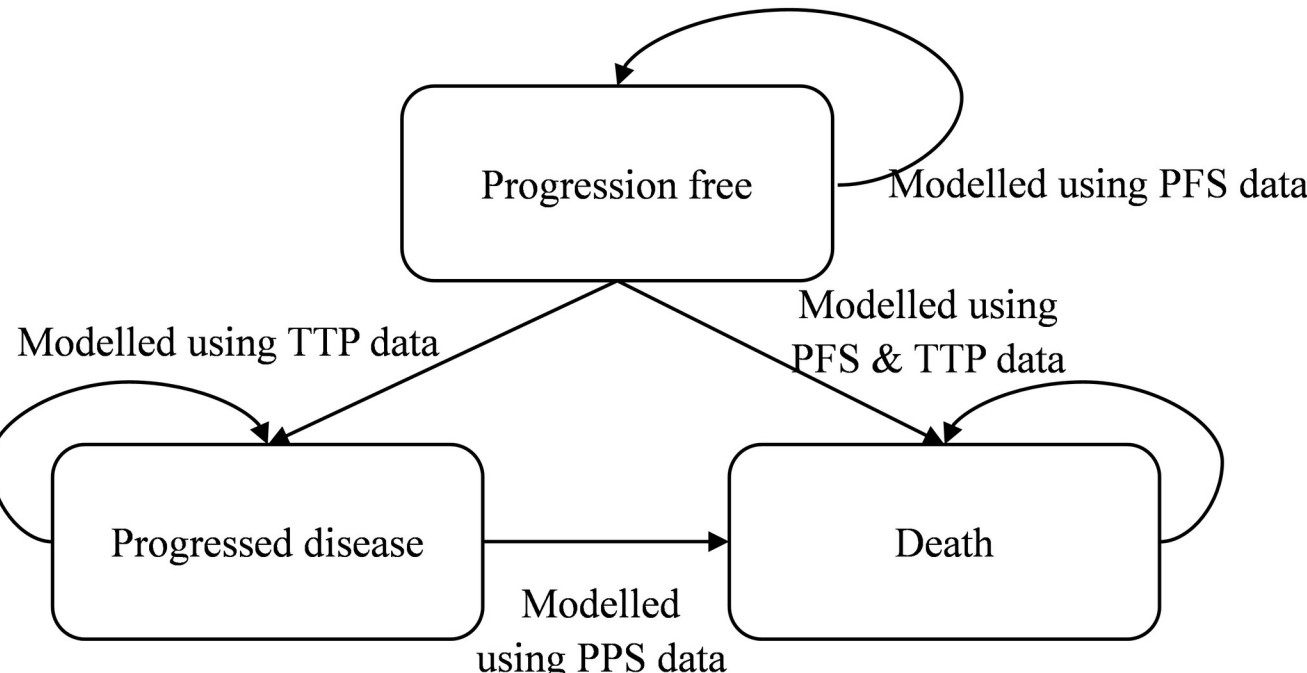

**Fig 1. Markov model structure.** The health-state structure was published previously as part of the UK National Institute for Care and Excellence single technology appraisal committee papers [22]. PFS, progression free survival; TTP, time to progression; PPS, post-progression survival.

transition probability of patients remaining in PF state was deduced based on progression free survival (PFS) data; the transition probability of patients from PF state to PD was derived from time to progression (TTP) data; and post-progression survival (PPS) data was used to deduce the transition probability of patients from PD state to death state. Since the model simulated time horizon was out of the follow-up period, the survival function, $S(t)$, was obtain and extrapolated by fitting exponential, Weibull, Log-normal, Log-logistic, Gompertz and generalized gamma distributions to the survival data, by selecting the best fit distribution based on Akaike Information Criterion (AIC) and Bayesian Information Criterion (BIC). The results showed that generalized gamma distribution represented the best goodness of fit to both TTP data and PFS data from patients in these two groups; and Log-normal distribution was the best fit distribution for PPS data from overall population. Survival function of generalized gamma distribution is $S(t) = 1 - \Gamma\left[k^{-2}e^{\left(\frac{k[\log(t)-\beta]}{\sigma}\right)}; k^{-2}\right]$; while survival function of Log-normal distribution is given as $S(t) = 1 - \int_{-\infty}^{(\ln t - \mu)/\sigma} f(x)dx$ (Table 1) [17, 18].

## Cost

Direct medical costs which consisted of costs of drug acquisition, subsequent treatment, medical resource utilization (including medical service, inspections and terminal care) and adverse event treatment were considered in this model.

The retail price of durvalumab in Chinese public hospitals was applied with a recommended dose of 10 mg/kg q2w (every 2 weeks) through intravenous infusion for more than 60 minutes until PD or unacceptable toxicity, and the treatment duration should not exceed 12 months. For the subsequent treatment interventions, the costs of immunotherapy, targeted therapy and chemotherapy regimen were derived from the bid price of drugs; the cost of radiotherapy regimen (including positioning, mold and radiotherapy) was consulted from clinical expert survey. The costs of medical service, inspections and terminal care were calculated based on the unit price of each item, patient proportion and frequency with associated parameters consulted from clinical expert survey. The incidence of adverse events in durvalumab group and BSC group was obtained from Asian subpopulation analysis in PACIFIC study, and the treatment cost for each adverse event was consulted from clinical expert survey.

For the purpose of improving the accessibility and standardization of immunotherapy for patients with lung cancer in China and reducing the economic burden of patients, China Primary Health Care Foundation launched a Patient Assistance Program (PAP) for durvalumab injection: patient who has used two cycles of medicines at his or her own cost in the first stage may obtain two cycles of medicine assistance; for the patient who has used four cycles of medicines at his or her own cost in the second stage, on the premise that the treatment for the

**Table 1. Parameters of survival curve.**

| TTP Survival curve | $\beta$ | $\sigma$ | $k$ |
|---|---|---|---|
| Durvalumab group | 0.6323 | 0.6480 | 7.3951 |
| BSC group | 0.6384 | 0.5739 | 4.2698 |
| **PFS survival curve** | $\beta$ | $\sigma$ | $k$ |
| Durvalumab group | 0.6478 | 0.6448 | 6.9943 |
| BSC group | 0.6274 | 0.5416 | 4.0697 |
| **PPS Survival curve** | $\mu$ | $\sigma$ | — |
| Both Durvalumab and BSC group | 3.0448 | 1.1876 | — |

BSC, best supportive care.

patient is progression-free and that the patient continues to benefit from such treatment and meets the eligibility criteria, the Program will continue to offer medicine assistance to the patient till disease progression. In this study, the cost of durvalumab in the scenario of PAP was converted according to the assistance plan.

## Utility

In the Pacific Study, health-related quality of life data were collected using the EuroQoL 5-dimension 5-level (EQ-5D-5L) questionnaire. In this model, we built a mixed effect model of utility values based on the patient-level data from the sub-population of Asia ethnicity of PACIFIC study, which showed that the utility value of patients in the PF state was 0.901, and the utility value in the PD state is 0.863. Adverse event related disabilities were not considered because the impact of AEs on health-related quality of life was assumed to be accounted for in patients' health states utilities.

## Base case analysis and sensitivity analysis

Results of patients with PAP and without PAP were calculated respectively in the base case scenario. The outcomes of this model included life year (LY), quality adjusted life year (QALY) and direct medical cost. A 5% discount rate for cost and QALY was applied according to recommendations of "China Guidelines for Pharmacoeconomic Evaluation" to calculate the incremental cost-effectiveness ratio (ICER) and compare with the willing to pay (WTP) value for cost-effectiveness analysis [19]. Based on the threshold of three times GDP per capita for judgment proposed by WHO, the three times GDP per capita of China in 2019 (212,676 yuan) was set as the threshold in this study [20].

One-way sensitivity analysis (OWSA) was used to evaluate the effect of each input parameter value variation within the 95% confidence interval (CI) on ICER in the base case scenario. When SE was not available, a 10% SE was assumed. The results of OWSA was presented in tornado diagram. Additionally, Monte Carlo simulation was used to carry out probabilistic sensitivity analysis (PSA) in which sampling was repeated for 1000 times based on the distribution specified for each parameter, with ICER value of each sampling calculated. PSA results were presented in cost-effectiveness plane scatter plot and cost-effectiveness acceptable curve (CEAC). The distribution of each parameters was fitted according to standard practice or guidelines (Table 2).

## Results

### Base case analysis

Discounted results for Durvalumab following concurrent chemoradiotherapy (cCRT) versus BSC are presented in Table 3. The life years of durvalumab group (7.39 years) were 2.60 years longer than those of BSC group (4.79 years); and 2.37 more QALYs gained in durvalumab group (6.61 QALYs) than those in BSC group (4.24 QALYs). The QALY difference between these two groups was 2.91 QALYs in the PF state and 0.54 QALYs in the PD state.

When the PAP was not considered, the lifetime cost for patients in durvalumab group was 459,027.13 yuan higher than that in BSC group, with an ICER of 193,898.00 yuan/QALY gained; when PAP was considered, the lifetime cost for patients in durvalumab group was 109,119.18 yuan higher than that in BSC group with an ICER value of 46,093.12 yuan/QALY gained (Table 3). The ICER in both cases were lower than the threshold of 3 times GDP per capita of China in 2019 (212,676 yuan).

**Table 2. Model parameters and distributions.**

| Parameter | Base case value | Standard error (SE) | Range | | Distribution | Source |
|---|---|---|---|---|---|---|
| | | | Lower limit | Upper limit | | |
| Cost (unit:¥) | | | | | | |
| Durvalumab (120 mg/vial) | 6066 | 606.60 | 4935.54 | 7311.29 | Gamma | yaozh.com |
| Durvalumab (500mg/vial) | 18088 | 1808.80 | 14717.12 | 21801.28 | Gamma | yaozh.com |
| Nivolumab (40 mg/vial) | 4587 | 458.70 | 3732.17 | 5528.66 | Gamma | yaozh.com |
| Nivolumab (100mg/vial) | 9250 | 925.00 | 7526.17 | 11148.93 | Gamma | yaozh.com |
| Pembrolizumab (100 mg/vial) | 17918 | 1791.80 | 14578.80 | 21596.38 | Gamma | yaozh.com |
| Docetaxel (20 mg/vial) | 302.4 | 30.24 | 246.04 | 364.48 | Gamma | yaozh.com |
| Vinorelbine (10 mg/vial) | 126.9 | 12.69 | 103.25 | 152.95 | Gamma | yaozh.com |
| Erlotinib (0.15 g*7 tablets/box) | 1275.75 | 127.58 | 1038.00 | 1537.65 | Gamma | yaozh.com |
| Crizotinib (0.25 g*60 tablets/box) | 15600 | 1560.00 | 12692.78 | 18802.52 | Gamma | yaozh.com |
| Afatinib (40 mg*7 tablets/box) | 1400 | 140.00 | 1139.10 | 1687.41 | Gamma | yaozh.com |
| Gemcitabine (1 g/vial) | 787.72 | 78.77 | 640.92 | 949.43 | Gamma | yaozh.com |
| Carboplatin (0.1 g/vial) | 53.9 | 5.39 | 43.86 | 64.97 | Gamma | yaozh.com |
| Cisplatin (30 mg/vial) | 19.15 | 1.92 | 15.58 | 23.08 | Gamma | yaozh.com |
| Paclitaxel (0.1 g/vial) | 533.43 | 53.34 | 434.02 | 642.94 | Gamma | yaozh.com |
| Pemetrexed (0.5 g/vial) | 2776.97 | 277.70 | 2259.45 | 3347.05 | Gamma | yaozh.com |
| 3D-CRT | 30000 | 3000.00 | 24409.20 | 36158.68 | Gamma | Expert consultation |
| IMRT | 50000 | 5000.00 | 40682.00 | 60264.47 | Gamma | Expert consultation |
| IG-IMRT | 80000 | 8000.00 | 65091.19 | 96423.16 | Gamma | Expert consultation |
| TOMO- tomography radiotherapy | 100000 | 10000.00 | 81363.99 | 120528.95 | Gamma | Expert consultation |
| Utility value | | | | | | |
| Utility value in PF state | 0.901 | 0.009 | 0.883 | 0.918 | Beta | PACIFIC study |
| Utility value in PD state | 0.863 | 0.009 | 0.845 | 0.880 | Beta | PACIFIC study |
| Others | | | | | | |
| Discount rate | 5% | —— | 0% | 8% | Uniform | [18] |
| Patient age | 62.90 | 0.34 | 62.24 | 63.56 | Normal | PACIFIC study |
| Weight | 61.2 | 0.74 | | | Fixed | PACIFIC study |

3D-CRT: 3D conformal radiotherapy; IMRT: Intensity modulated radiotherapy; IG-IMRT: Image-guided intensity-modulated radiotherapy.

**Table 3. Base case results: Discounted cost-effectiveness of durvalumab.**

| | | Durvalumab | BSC | Incremental |
|---|---|---|---|---|
| Lys | PF | 5.96 | 2.74 | 3.23 |
| | PD | 1.43 | 2.06 | -0.63 |
| | Total | 7.39 | 4.79 | 2.60 |
| QALYs | PF | 5.37 | 2.46 | 2.91 |
| | PD | 1.24 | 1.78 | -0.54 |
| | Total | 6.61 | 4.24 | 2.37 |
| Cost | Without PAP | 707,268.14 | 248,241.01 | 459,027.13 |
| | With PAP | 357,360.19 | 248,241.01 | 109,119.18 |
| ICER | Without PAP | | | 193,898.00 |
| | With PAP | | | 46,093.12 |

LY, life year; QALY, quality adjusted life years; PAP, Patient Assistance Program; BSC, best supportive care; ICER, incremental cost-effectiveness ratio; PF, progression free; PD, progressed disease.

The relatively high drug costs of Durvalumab were partly offset by lower subsequent therapy costs and terminal care costs. Durvalumab was also associated with higher health care resource utilization costs because patients stay longer in the PF state.

## Sensitivity analysis

One-way sensitivity analysis for the case without PAP of drugs showed that the unit price of durvalumab 500 mg and durvalumab 120 mg, and utility value in PF state were the main influential factors for ICER (Fig 2). Tornado diagram for the case with free drugs was not presented here due to the length limit of paper.

The results of probabilistic sensitivity analysis in the case without PAP of drugs showed that, the average incremental cost of 1000 Monte Carlo simulations was 461,454.93 yuan with an average of 2.20 incremental QALYs gained, yielding an ICER of 209,418.93 yuan/QALY gained similar to that in the basic analysis, which indicated that the model has a good robustness. From scatter plot of cost-effectiveness plane (Fig 3), 53.2% scatter dots can be seen below the line of willing payment, which suggested that the probability of cost-effectiveness for durvalumab is 53.2% based on the threshold defined in this study. The cost-effectiveness acceptable curve (Fig 4) showed that the probability of durvalumab being cost-effective increased with the average social WTP. When WTP was 200,000 yuan in the case without PAP, the probability of durvalumab being cost-effective is higher than BSC.

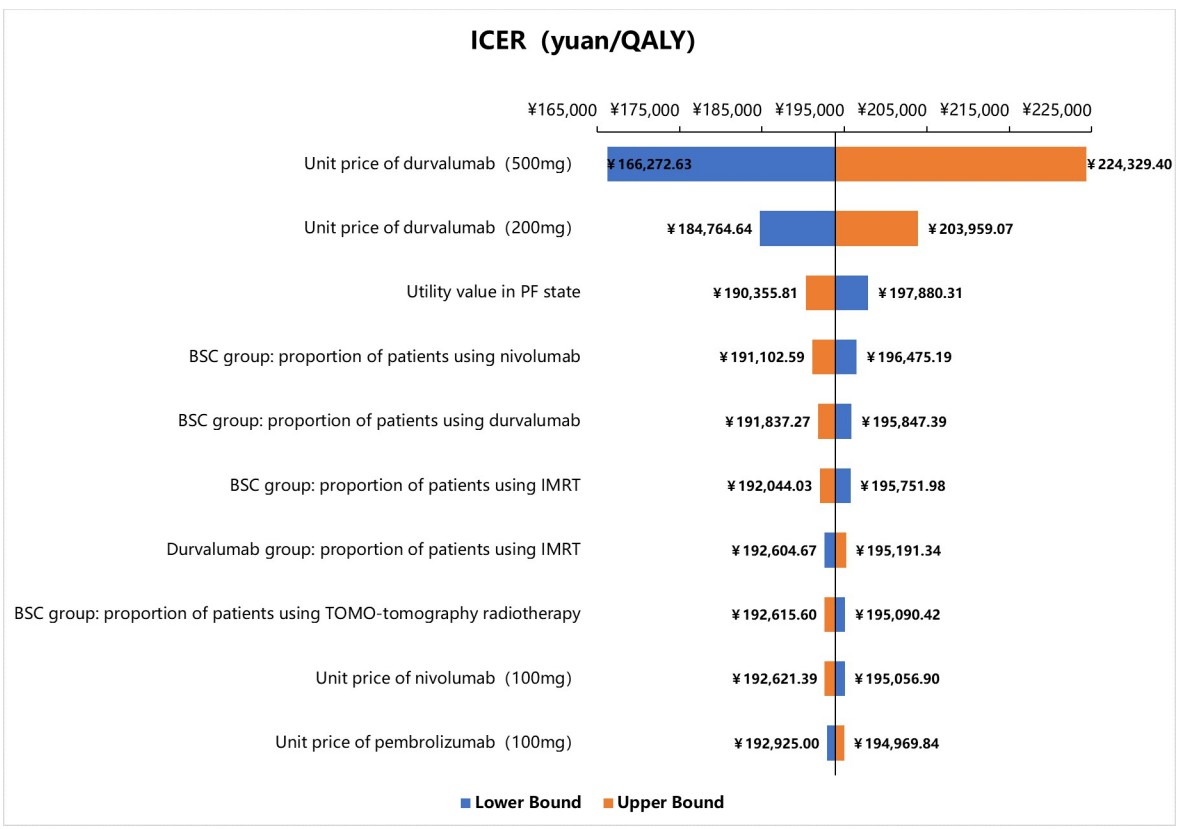

**Fig 2. Tornado diagram of one-way sensitivity analysis.** PF, progression free; IMRT, Intensity modulated radiotherapy; BSC, best supportive care.

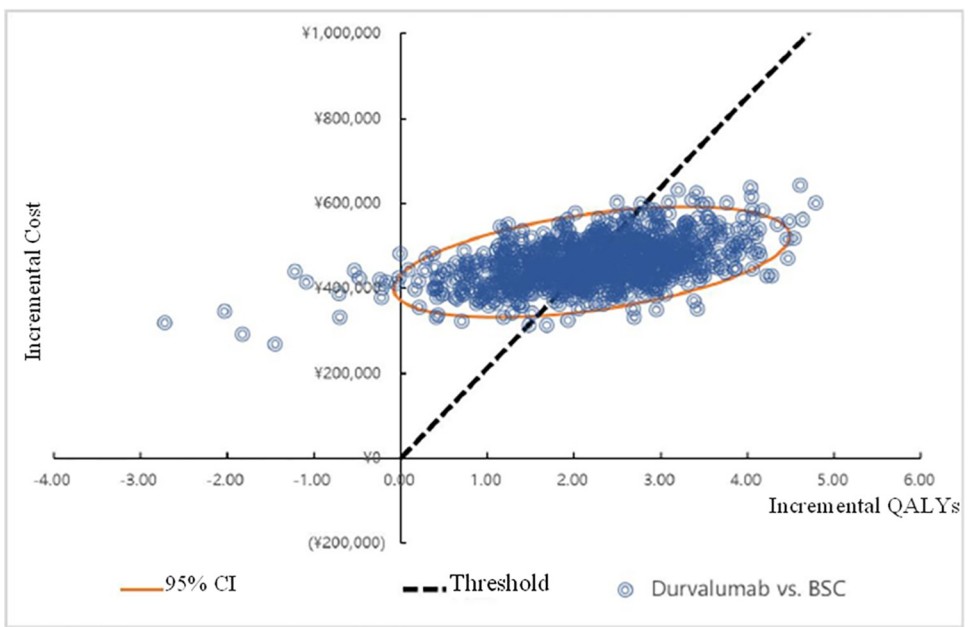

**Fig 3. Cost-effectiveness plane scatter plot.**

## Discussion

We constructed a state transition model based on the latest patient-level data from Asian ethnicity population in PACIFIC study and consultation data from clinical expert questionnaire to analyse the cost-effectiveness of durvalumab versus BSC in the treatment of patients with unresectable stage III NSCLC in China. In basic analysis, the life years were prolonged 2.60 years and 2.37 more QALYs gained in durvalumab group compared with those in BSC group. The patient's lifetime cost was 459,027.13 yuan and 109,119.18 yuan higher in the case without

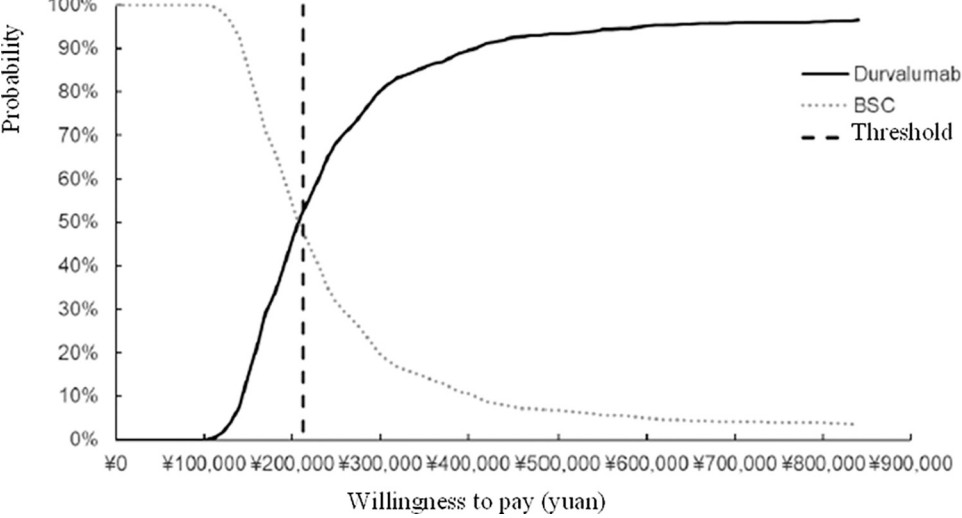

**Fig 4. Cost-effectiveness acceptable curve.** The acceptability curve shows the probability of a treatment being cost-effective over a range of willingness-to-pay thresholds. The curve shows that BSC is the most likely to be cost-effective until a willingness-to-pay threshold of 200,000 yuan is reached, after which durvalumab is most likely to be cost-effective.

**Table 4. Comparison of cost-effectiveness studies on durvalumab.**

| | US study [11] | US study [12] | US study [13] | Switzerland study [14] | Italy study [15] | UK study [16] | Present study* |
|---|---|---|---|---|---|---|---|
| **Basic information of model** | | | | | | | |
| Study perspective | Society | Healthcare payers | Medicare and commercial insurance perspectives | Healthcare payers | Healthcare payers | Healthcare payers | Healthcare system |
| Model type | Microsimulation model | Markov model | Semi-Markov model | Markov model | Decision tree + Markov model | State transition model | State transition model |
| Target population | Pacific ITT | Pacific ITT | Pacific ITT | Pacific ITT & PD-L1≥1% | Pacific PD-L1≥1% | Pacific PD-L1≥1% | Pacific Asian ethnicity |
| Time horizon | 5 years | Lifetime | 30 years | 10 years | Decision tree model: 1 year; Markov model: 40 years | 40 years (lifetime) | Lifetime |
| Cycle period | 1 months | 1 months | 2 weeks for the first 12 months and 4 weeks thereafter | 1 months | 1 months | 2 weeks for the first 12 months and 4 weeks thereafter | 2 weeks for the first 12 months and 4 weeks thereafter |
| Discount rate | 3% | 3% | 3% | 3% | 3% | 3.5% | 5% |
| Utility value | PF: 0.79; PD: 0.76 | PF: 0.791 [a], 0.809 [b]; first progression: 0.653; second progression: 0.473 | NA | PF in Year 1: 0.69 [a]; PF after Year 1 [ab]: 0.71; PD: 0.65 | Stage III, PF: 0.810; stage IV, first-line, PF: 0.710; stage IV, first-line, PD: 0.670; stage IV, second-line, PF: 0.740; stage IV, first-line, PD: 0.590. | PF: 0.810; PD: 0.776 | PF: 0.901; PD: 0.863 |
| **Main results** | | | | | | | |
| Life year | 3.87 vs 3.65 | 4.85 vs 3.51 | 6.08 vs 4.14 | 4.49 vs 3.49 | 3.47 vs 3.31 | NA | 7.39 vs 4.79 |
| QALYs | 2.57 vs 2.34 | 3.13 vs 2.12 | 5.13 vs 3.47 | 2.93 vs 2.17 | 2.73 vs 2.50 | △2.51 | 6.61 vs 4.24 |
| Cost | $201563 vs $185944 | $336410 vs $195324 | $206818 vs $115395 | CHF 180206 vs CHF 112966 | €59860 vs €48840 | NA | ¥707268 vs ¥248241 |
| | | | $244582 vs $142524 | | | | |
| ICER | $67421/QALY | $139689/QALY | $55285/QALY $61111/QALY | CHF 88703/ QALY | €42322/QALY | £22665/QALY | ¥193898/QALY |
| Threshold | $100000 | $150000 | $100000 | CHF 100000 | €16372 | £30000 | ¥212676 |

PF, progression free; PD, progression disease; ICER, incremental cost-effectiveness ratio

△, indicates the increment; NA, indicates that no data reported for this item; ITT: intention to treat a the utility value of durvalumab group. b the utility value of BSC

* only results without PAP were reported.

or with PAP of drugs respectively, resulting in ICER values of 193898.00 yuan/QALY gained and 46,093.12 yuan/QALY gained, respectively. The results revealed that durvalumab prolonged the life years, improved the quality of life, while increased medical costs. Based on the threshold of three times GDP per capita for judgment proposed by WHO, durvalumab demonstrated a comparative advantage of cost-effectiveness regardless of PAP.

Till now, there has not been any economic research on durvalumab in China, and a total of 6 economic studies on durvalumab versus BSC have been retrieved out of China, with 3 from US and 1 each from Switzerland, Italy and UK [11–16]. The clinical data of these six studies were obtained from PACIFIC study with the same control group of BSC (Table 4). These studies focused on the whole population or patients with PD-L1≥1%, which are quite different from our study because the clinical and utility data were obtained from patient level data of Asian ethnicity. These studies were varied in the target population, study perspectives, models, clinical background and resources of cost data, hence different conclusions were drawn. The

results of three studies from United States, one study from Switzerland and one study from UK showed that the use of durvalumab was cost-effective. One study from Italy perspective showed that Durvalumab was not cost effective at listing price, but if a discount was offered, Durvalumab would be a cost-effective treatment. Anyway, these five studies all presented the cost-effectiveness of durvalumab from different perspectives, providing more comprehensive implications for decision makers.

It is very important to explore the threshold for pharmacoeconomic evaluation of advanced cancer treatment. At present, there is no unified standard on the evaluation of QALYs in China, so the threshold of this study was defined as 3 times GDP per capita of China in 2019 based on the "China Guidelines for Pharmacoeconomic Evaluation" and referred to the WHO recommendation of using disability-adjusted life years as outcome measure on economic evaluation: if GDP per capita < ICER < 3 times GDP per capita, then increased cost is acceptable [19]. The ICER for Durvalumab versus BSC are 193,898.00 RMB/QALY and 46,093.12 RMB/QALY without or with PAP respectively, which are about 2.74 and 0.65 times as much as per capita GDP in 2019. For other countries globally, National institute for Health and Care Excellence (NICE) in UK proposed a threshold of ≤ 30,000 pounds higher than the general standard for drugs qualified for relevant requirements and capable to prolong the survival of patients at end stage; an US study on the threshold of WTP for new anti-cancer drugs showed that the average acceptable cost is $300,000 [21]. Therefore, further discussion is required for the setting standard of threshold in economic evaluation studies based on Chinese population.

This study has following limitations: First, due to the immaturity of survival data including TTP, PFS and PPS from patients in PACIFIC study, the survival curves need to be extrapolated in our model, therefore, it is still necessary to verify these results using mature survival curve data later though the influence of uncertainty for each survival curve on the analysis results has been discussed in probability sensitivity analysis. Second, durvalumab injection was just launched in Chinese market since December 2019, it is difficult to obtain representative real-world data to support the cost parameters required in the model. So, it is necessary to carry out more real-world studies on durvalumab to verify the parameters consulted from clinical expert questionnaire in the model.

## Conclusions

Our study shows that Durvalumab following cCRT is a cost-effective treatment option compared with BSC in patients with stage III non-small cell lung cancer from the healthcare system perspective in China.

## Supporting information

**S1 Fig. KM data, active curve and outcome.**
(PDF)

**S1 Table. The original clinical efficacy data from the PACIFIC study.**
(PDF)

## Acknowledgments

Thanks are due to Cancer Hospital Chinese Academy of Medical Sciences, Peking University People's Hospital, Beijing Cancer Hospital, Fudan University Shanghai Cancer Center and The First Affiliate Hospital of GUANGZHOU Medical University for valuable discussion. The authors thank Peter Elroy and his colleagues (BresMed Health Solutions, Sheffield, UK) for the global model development and the utility analyses.

## Author Contributions

**Investigation:** Jinyu Chen, Hanxue Jia.

**Methodology:** Jinyu Chen.

**Software:** Min Zheng.

**Writing – original draft:** Xiaotong Jiang, Jinyu Chen, Min Zheng, Hanxue Jia.

**Writing – review & editing:** Jinyu Chen.

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
