## [Decision Letter · Decision Letter 0]

24 Nov 2021

PONE-D-21-27815COST-EFFECTIVENESS ANALYSIS OF DURVALUMAB AS A MAINTENANCE TREATMENT FOR PATIENTS WITH LOCALLY ADVANCED, UNRESECTABLE, STAGE III NSCLC IN CHINAPLOS ONE

Dear Dr. Chen,

Thank you for submitting your manuscript to PLOS ONE. After careful consideration, we feel that it has merit but does not fully meet PLOS ONE’s publication criteria as it currently stands. Therefore, we invite you to submit a revised version of the manuscript that addresses the points raised during the review process.

We look forward to receiving your revised manuscript.

Kind regards,

Jianxin Xue

Academic Editor

PLOS ONE

Journal Requirements:

2. We note you have included a table to which you do not refer in the text of your manuscript. Please ensure that you refer to Table 2 in your text; if accepted, production will need this reference to link the reader to the Table

Reviewers' comments:

Reviewer's Responses to Questions

**Comments to the Author**

1. Is the manuscript technically sound, and do the data support the conclusions?

Reviewer #1: Yes

Reviewer #2: Partly

2. Has the statistical analysis been performed appropriately and rigorously? 

Reviewer #1: Yes

Reviewer #2: Yes

3. Have the authors made all data underlying the findings in their manuscript fully available?

Reviewer #1: Yes

Reviewer #2: Yes

4. Is the manuscript presented in an intelligible fashion and written in standard English?

Reviewer #1: Yes

Reviewer #2: Yes

5. Review Comments to the Author

Reviewer #1: The study comments on the UK cost-effectiveness perspective of Witlox W et al stating that "Durvalumab was not cost-effective because the unmatured of survival data". I think it is worthy to cite the updated perspective after 4 years survival in the report: Dunlop, W., van Keep, M., Elroy, P. et al. Cost Effectiveness of Durvalumab in Unresectable Stage III NSCLC: 4-Year Survival Update and Model Validation from a UK Healthcare Perspective. PharmacoEconomics Open (2021). https://doi.org/10.1007/s41669-021-00301-7

China is a big market. It would be helpful to clarify the way of curation of drugs. Is it a free market with governmental licensing or a central curation authority able to negotiate price discounts, that would affect cost-effectiveness evaluations by the power of monopsony.

Reviewer #2: The study shows that Durvalumab following cCRT is a cost-effective treatment option compared with BSC in patients with stage Ⅲ non-small cell lung cancer from the healthcare system perspective in China..

The paper is solid from the statistical point of view, but the authors should explain in more detail the importance of comparing these two treatments beyond costs.Has Durvalumab following cCRT better results than BSC?

6. PLOS authors have the option to publish the peer review history of their article (what does this mean?). If published, this will include your full peer review and any attached files.

Reviewer #1: **Yes: **Salah-Eldin Abdelmoneim

Reviewer #2: No

---

## [Author Response · Author response to Decision Letter 0]

1 Feb 2022

Replies to the reviewers’ comments:

Reviewer #1:

1. The study comments on the UK cost-effectiveness perspective of Witlox W et al stating that "Durvalumab was not cost-effective because the unmatured of survival data". I think it is worthy to cite the updated perspective after 4 years survival in the report: Dunlop, W., van Keep, M., Elroy, P. et al. Cost Effectiveness of Durvalumab in Unresectable Stage III NSCLC: 4-Year Survival Update and Model Validation from a UK Healthcare Perspective. PharmacoEconomics Open (2021). https://doi.org/10.1007/s41669-021-00301-7.

Response: We have replaced the reference and updated the values in Table 4 as well as the descriptions in the text.

2. China is a big market. It would be helpful to clarify the way of curation of drugs. Is it a free market with governmental licensing or a central curation authority able to negotiate price discounts, that would affect cost-effectiveness evaluations by the power of monopsony.

Response: It is a central curation authority able to negotiate price discounts. The National Healthcare Security Administration, NHSA manages the price negotiation, known as the National Reimbursement Drug List (NRDL). Drugs that make it on this list are covered by national health insurance, available to nearly all Chinese citizens today.

Reviewer #2:

1. The paper is solid from the statistical point of view, but the authors should explain in more detail the importance of comparing these two treatments beyond costs. Has Durvalumab following cCRT better results than BSC?

Response: The life years of durvalumab group (7.39 years) were 2.6 years longer than those of BSC group (4.79 years); and more QALYs gained in durvalumab group (6.61 QALYs) than those in BSC group (4.24 QALYs).

---

## [Decision Letter · Decision Letter 1]

6 Jun 2022

COST-EFFECTIVENESS ANALYSIS OF DURVALUMAB AS A MAINTENANCE TREATMENT FOR PATIENTS WITH LOCALLY ADVANCED, UNRESECTABLE, STAGE III NSCLC IN CHINA

PONE-D-21-27815R1

Dear Dr. Chen,

We’re pleased to inform you that your manuscript has been judged scientifically suitable for publication and will be formally accepted for publication once it meets all outstanding technical requirements.

Kind regards,

Jun Hyeok Lim, M.D.

Academic Editor

PLOS ONE

Additional Editor Comments (optional):

Reviewers' comments:

Reviewer's Responses to Questions

**Comments to the Author**

1. If the authors have adequately addressed your comments raised in a previous round of review and you feel that this manuscript is now acceptable for publication, you may indicate that here to bypass the “Comments to the Author” section, enter your conflict of interest statement in the “Confidential to Editor” section, and submit your "Accept" recommendation.

Reviewer #2: All comments have been addressed

2. Is the manuscript technically sound, and do the data support the conclusions?

Reviewer #2: Yes

3. Has the statistical analysis been performed appropriately and rigorously? 

Reviewer #2: Yes

4. Have the authors made all data underlying the findings in their manuscript fully available?

Reviewer #2: Yes

5. Is the manuscript presented in an intelligible fashion and written in standard English?

Reviewer #2: Yes

6. Review Comments to the Author

Reviewer #2: The authors have addressed all the comments.This paper is interesting and is now in an position to be published

7. PLOS authors have the option to publish the peer review history of their article (what does this mean?). If published, this will include your full peer review and any attached files.

Reviewer #2: No

---

## [Editor Report · Acceptance letter]

13 Jun 2022

PONE-D-21-27815R1 

COST-EFFECTIVENESS ANALYSIS OF DURVALUMAB AS A MAINTENANCE TREATMENT FOR PATIENTS WITH LOCALLY ADVANCED, UNRESECTABLE, STAGE Ⅲ NSCLC IN CHINA 

Dear Dr. Chen:

I'm pleased to inform you that your manuscript has been deemed suitable for publication in PLOS ONE. Congratulations! Your manuscript is now with our production department. 

Kind regards, 

on behalf of

Dr. Jun Hyeok Lim 

Academic Editor

PLOS ONE